# Trip-Related Fall Risk Prediction Based on Gait Pattern in Healthy Older Adults: A Machine-Learning Approach

**DOI:** 10.3390/s23125536

**Published:** 2023-06-13

**Authors:** Shuaijie Wang, Tuan Khang Nguyen, Tanvi Bhatt

**Affiliations:** 1Department of Physical Therapy, University of Illinois at Chicago, Chicago, IL 60612, USA; sjwang4@uic.edu; 2Department of Computer Science, University of Illinois at Chicago, Chicago, IL 60607, USA; tnguy272@uic.edu

**Keywords:** trip, fall assessment, ensemble classification, gait characteristics

## Abstract

Trip perturbations are proposed to be a leading cause of falls in older adults. To prevent trip-falls, trip-related fall risk should be assessed and subsequent task-specific interventions improving recovery skills from forward balance loss should be provided to the individuals at risk of trip-fall. Therefore, this study aimed to develop trip-related fall risk prediction models from one’s regular gait pattern using machine-learning approaches. A total of 298 older adults (≥60 years) who experienced a novel obstacle-induced trip perturbation in the laboratory were included in this study. Their trip outcomes were classified into three classes: no-falls (*n* = 192), falls with lowering strategy (L-fall, *n* = 84), and falls with elevating strategy (E-fall, *n* = 22). A total of 40 gait characteristics, which could potentially affect trip outcomes, were calculated in the regular walking trial before the trip trial. The top 50% of features (*n* = 20) were selected to train the prediction models using a relief-based feature selection algorithm, and an ensemble classification model was selected and trained with different numbers of features (1–20). A ten-times five-fold stratified method was utilized for cross-validation. Our results suggested that the trained models with different feature numbers showed an overall accuracy between 67% and 89% at the default cutoff and between 70% and 94% at the optimal cutoff. The prediction accuracy roughly increased along with the number of features. Among all the models, the one with 17 features could be considered the best model with the highest AUC of 0.96, and the model with 8 features could be considered the optimal model, which had a comparable AUC of 0.93 and fewer features. This study revealed that gait characteristics in regular walking could accurately predict the trip-related fall risk for healthy older adults, and the developed models could be a helpful assessment tool to identify the individuals at risk of trip-falls.

## 1. Introduction

Falls in older adults are a worldwide health concern [1] which can lead to serious medical problems [2,3], and over 90% of hip fractures are induced by falls in older adults in the United States. Trip perturbation, responsible for over 50% of outdoor falls [4], is proposed to be the leading cause of falls in older adults [5,6]. In order to decrease the likelihood of a fall incidence, providing prevention programs to high-risk older adults would be beneficial, especially to those with a high probability of trip-related falls. Various factors can contribute to falls, leading to different types of fall. In order to combat each unique fall type, it is imperative to utilize recovery strategies that cater to each individual possible fall; correspondingly, treatment strategies should be task-specific rather than generic to efficiently lower fall risk. Thus, an accurate assessment of one’s fall risk type would be a prerequisite to provide/develop specific interventions.

Previous studies have investigated how the kinematic factors display a considerable discrepancy between fallers and no-fallers in healthy older adults [7,8]; with this, response time from trip-onset to recovery foot touchdown was found to be a crucial component affecting trip outcomes (fall or not) for participants with foot-lowering strategies (post-perturbation tripping limb touchdown behind the obstacle) [9,10]. However, it is unconfirmed whether this factor could be parallelly utilized to identify the fallers with elevating strategies (post-perturbation tripping limb touchdown in front of the obstacle). Similarly, another study reported that recovery stepping behavior (single or multiple recovery steps) also showed a significant difference between fallers and no-fallers [11], and this might be used as a potential predictor of future falls in older adults. Despite the lower prediction accuracy of fall risk, these reactive factors could only be measured with the assistance of clinicians/physicians using perturbation-trigger devices, which limited their application in clinical and home settings.

Gait characteristics, which are easily assessed using wearable sensors (i.e., IMU sensors), have a significant correlation with physical function, muscle power, and dynamic ability [12,13,14,15]. Our previous studies have revealed that gait characteristics could predict slip-related fall risk in both heathy older adults and stroke patients [16,17], and many studies have proposed that gait characteristics could also be used as a predictor(s) to forecast the fall risk following a trip perturbation, such as foot clearance [18,19], step length, gait speed, trunk flexion angle, and velocity [20]. Based on step time and step length, Pavol et al. developed trip-related fall risk prediction models with a high prediction accuracy of 89.8% [21]. However, only 10 fallers were used to train their models and the recovery strategy (lowering or elevating) was not specified; the small sample size limited the statistical power and the generalization of their study. Furthermore, these studies only take single or dual factors into account for fall-risk prediction, possibly yielding inaccurate and unreliable results. It is imperative that fall-risk models consider the wide spectrum of factors leading to falls. However, by including increased dimensionality, redundancy, and irrelevant data, classification could increase demand and could even lead to a lower prediction accuracy.

Machine learning is a subset of artificial intelligence that uses large amounts of data to make outcome predictions [22], which could help to identify significant factors in high-dimensional data. Such a method is proven to be powerful and efficient, capable of producing a higher prediction accuracy compared to the traditional statistical methods. The purpose of this study was to develop prediction models to predict trip-related fallers with different recovery strategies using machine-learning methods. Along with this, the accuracy of models with a diverse number of features was compared to determine the optimal feature size for fall prediction.

## 2. Methods

### 2.1. Participants

A total of 298 community-dwelling older adults (age: 69.6 ± 6.33 years; weight: 75.2 ± 17.2 kg; height: 1.67 ± 0.1 m; gender: 128 M/170 F) were included in this study. All participants were screened via questionnaires before their participation. Exclusionary criteria included neurological, musculoskeletal, cardiopulmonary, or any other system disorders. All participants provided written informed consent, and this study was conducted according to the guidelines of the Declaration of Helsinki and approved by the Institutional Review Board of the University of Illinois at Chicago (IRB#: 2016-0887).

### 2.2. Experimental Setup

The trip perturbation was induced by an obstacle device, which consisted of a hinged metal plate (height: 8 cm; width: 27 cm; thickness: 0.5 cm) and a pair of electromagnets. The plate was locked in a flat position by a pair of electromagnets during regular walking, and the electromagnets were powered off in trip trials once the vertical ground reaction force under the unperturbed (right) limb exceeded 80% of participants’ body weight. The plate could reach its upright position in less than 150 ms after right foot touchdown (Figure 1a). The ground reaction force was detected by the force plates (AMTI, Newton, MA) installed beneath the right platform.

Participants experienced at least 25 regular walking trials followed by a trip trial (Figure 1b). For both regular walking and trip trials, all subjects were instructed to walk at their preferred (self-selected) speed and manner on the 7-m walkway. They were told “a trip may or may not occur during your walking”. Their starting position was adjusted during regular walking to ensure the upcoming trip obstacle consistently obstructed the participant’s left foot in the mid-to-late swing phase [23]. All the gait characteristics included in our study were calculated in the regular walking trial before the novel trip.

A full-body safety harness connected by shock-absorbing ropes to a loadcell was used to protect participants from falling and detect the harness-supported body weight (Transcell Technology Inc., Buffalo Grove, IL, USA). The loadcell was mounted to an overhead trolley on a track over the walkway. The harness enabled participants to walk freely while providing protection against body impact with the floor. Kinematics from a full-body marker set (31 retro-reflective markers) were recorded by an eight-camera motion capture system (Qualisys AB, Gothenburg, Sweden). Specifically, 26 markers were placed on the body, and another 5 markers were placed on the walkway and the trip plate (Figure 1c). Kinematic data were sampled at 120 Hz and synchronized with the force plate and loadcell data, which were collected at 600 Hz.

### 2.3. Balance Outcomes

The trip outcome was identified as a fall or no-fall based on whether the loadcell force was over 30% of the body weight [24], and then the trip outcomes were further unambiguously cross-checked with careful visual inspection of the video records. The fall trials were further divided into two groups based on the adopted recovery strategy (lowering or elevating) [10]: the lowering strategy consists of lowering the tripped limb behind the obstacle and taking a recovery step with the other limb, and the elevating strategy consists of taking a recovery step by lifting the tripped foot over the obstacle. Among all the 298 trip trials, 192 were identified as no-fall, 84 were identified as a fall with lowering strategy (L-fall), and the remaining 22 were identified as a fall with elevating strategy (E-fall).

### 2.4. Kinematic Measures

All of the spatiotemporal gait characteristics which could potentially affect trip outcomes were included in our study, including gait speed, trunk angle, center-of-mass (*COM*) state, step length, gait duration, toe clearance, swing time, and step time [21,25,26,27]. Considering that these gait characteristics are highly related to joint kinematics, the hip angle, knee angle, and foot angle for both limbs were also calculated at different time event. All of the measures were calculated in one gait cycle from the natural walking trial prior to the trip trial (Table 1). The gait cycle was from the unperturbed (right) foot touchdown (pre-TD) to its next touchdown; this gait cycle was selected as it contains a perturbed (left) foot liftoff (LO) and its touchdown (post-TD), which were immediately before and after the trip onset in the following trip trial.

Gait speed was calculated based on the *COM* velocity, and the *COM* kinematics were calculated using a 13-segment rigid body model with sex-dependent segmental inertial parameters using the position of the 3D markers. The average gait speed (in a gait cycle), maximum gait speed, and instantaneous gait speed at pre-TD, LO, and post-TD were included in this study. The *COM* state includes the *COM* position relative to the posterior BOS, *COM* velocity (*vCOM*), and margin of stability (*MOS*) calculated based on the former two variables as in the equation below, while the *g* and *l* in the equation represent the gravitational acceleration and the leg length [28]:$$MOS=COM-{\mathrm{BOS}}_{min}+vCOM/\sqrt{\left(g/l\right)}$$

Gait duration included the swing time (LO to post-TD) and step time (pre-TD to post-TD). Step length was calculated as the heel distance at pre-TD and post-TD. Toe clearance was calculated as the maximum toe height in the swing phase. Trunk angle was calculated as the angle between the trunk segment and horizontal plane: Θtrunk = atan2(shoulder_y_ − hip_y_, shoulder_x_ − hip_y_)

Hip angle was calculated as the thigh segment and the horizontal plane: Θhip = atan2(hip_y_ − knee_y_, hip_x_ − knee_x_)

Foot angle was calculated as the foot segment and the horizontal plane: Θfoot = atan2(toe_y_ − heel_y_, toe_x_ − heel_x_)

Knee angle was the angle between the thigh and shank segments: Θknee = atan2(knee_y_ − ankle_y_, knee_x_ − ankle_x_) + (180 − Θhip)

All the joint angles were calculated at LO and post-TD; in addition, the maximum angles for the perturbed limb were calculated in the swing phase, and the maximum trunk angle was calculated in the gait cycle. In total, 40 features were included in our study for fall-risk prediction model training. Here, the feature was defined as an individual kinematic measure used for model training, and a model was built based on a combination of features to provide predictions of trip outcome using machine-learning algorithms.

### 2.5. Machine Learning

Feature selection: We ranked the importance of 40 features using the ReliefF algorithm, and the top 50% (*n* = 20) features were selected to train the model [29]. The main target in the feature-selection stage was to eliminate all the redundant and highly correlated features and reduce the extracted feature space into a smaller subset that is highly distributive and informative to ensure a high classification performance of the prediction model.

Model selection: To determine the best performing algorithm, eight different algorithms were trained using selected features: discriminant analysis classifier (DAC), ensemble classification model (ECM), kernel, k-nearest neighbor model (KNN), linear, naive Bayes classifier (NBC), support vector machine classifier (SVM), and binary decision classification tree. The model selection (Fitcauto) function from MATLAB 2021a (MathWorks Inc., Natick, MA, USA) was used for the training and hyperparameter tuning of the models [30]. The ECM model (learner: bag, minleafsize: 5) which showed the best performance was selected [31].

Model training: The ECM model was then used to train the model with different numbers of features. We optimized the hyperparameters (NumLearningCycles and MaxNumSplits) using Bayesian optimization. The model was firstly trained with all 20 features, and then we estimated the features’ importance using the predictorImportance function. The feature with the lowest score was removed from the model until only one feature remained. Thus, we obtained 20 models with different features.

Cross-validation: A 5-fold stratified method was used to split the data into a training set (80%) and a test set (20%). Each model with the same features was trained 10 times to embrace randomness during machine learning. The flow chart can be seen in Figure 2.

Mixed-model training: Demographic features were also considered related to the fall risk in older adults [32]; hence, we re-ran the model training steps (from feature selection to cross-validation) by adding demographic features (age, gender, height, and weight) to ascertain whether the inclusion of demographics could further improve the accuracy of the model.

### 2.6. Statistical Analysis

To evaluate the performance of each model, the receiver operating characteristic curve (ROC) for trip-outcome prediction (no-fall, E-fall, or L-fall) was firstly calculated. Next, the prediction accuracy (true positive rate) for each class and the overall prediction accuracy were calculated for each model at the default cutoff (0.5) and the optimal cutoff: at which the sum of the true positive rate and false negative rate is at a maximum. The area under the curve (AUC) is a robust metric of a model’s performance; hence, the AUC of each model was also calculated for each test set (*n* = 5 fold × 10 times), and a one-way ANOVA was conducted to compare the difference in AUC among these models with different number of features. Based on the value of the AUC, we defined the model with the maximum AUC as the best model, and the model with the fewest features and a comparable AUC (>0.95 × maximum AUC) as the optimal model. A post-hoc paired t-test was then used to compare each pair of models. All statistical analysis was performed using MATLAB 2021a.

## 3. Results

The trained models with different feature numbers showed an overall accuracy between 67% and 89% at the default cutoff (Table 2), and the prediction accuracy roughly increased along with the number of features (Figure 3). Among all the models, the one with 17 features showed the best prediction accuracy (88.9%). Specifically, 99.8% of the no-falls and 83% of the L-falls could be predicted, while only 17.8% of E-falls were accurately predicted. With a single feature as the predictor, the prediction accuracy for both types of falls was extremely low (<19% for both).

At the optimal cutoff, almost all 20 models showed a higher overall accuracy (70–94%) compared to the accuracy at the default cutoff, and the prediction accuracy for L-falls was greatly improved by at least 15% for all of the models. Take the model with 17 features as an example: 95.3% of the no-falls, 98.7% of the L-falls, and 68.2% of E-falls were accurately predicted. Among all the gait characteristics, *COM* velocity (Features 1 and 3 in Table 3) and joint angles of the perturbed limb (Features 2, 4, and 6) could be considered major contributors to the trip outcome prediction. Along with unperturbed foot angle at post-TD and toe clearance of the perturbed limb, over 80% of the trip outcomes could be predicted (model with 7 features in Table 3).

The one-way ANOVA results showed that the number of features has a significant effect on the AUC and accuracy (*p* < 0.001 for all). The model with 17 features could be considered the best model, with an AUC of 0.96 (Figure 4), and the model with 8 features could be considered the optimal model, with a comparable AUC (0.93 > 95% of the max AUC) achieved while removing over half of the features.

The models developed based on both gait characteristics and demographics showed no difference from those based only on gait characteristics, as none of the demographics were selected by the ReliefF algorithm, suggesting that these features were less important for trip-related fall risk training compared to the gait characteristics.

## 4. Discussion

This study focused on developing trip-related fall risk prediction models based on gait characteristics for healthy older adults, and the results indicated that there is a strong association between trip outcomes and gait patterns. While looking solely at a single gait feature, about 70% of trip outcomes could be predicted. Adding more relevant attributes led to an increase of 15% in prediction accuracy, with over 85% of falls predicted (models with 8–20 features in Table 2 and Table 3). The model seemed to reach a plateau when the prediction accuracy reached 94%, its accuracy stopped increasing when adding more features.

Our results are consistent with previous findings that gait speed and trunk angle are two of the key features affecting the likelihood of falls following a trip [21,26], and the fallers were reported to have a higher gait speed (average *COM* velocity) and larger trunk flexion compared to the no-fallers. In our ECM model, both the instantaneous and average *COM* velocity were important predictors (Rank 1 and 3 in Table 2), and the maximum trunk flexion and trunk angle at post-TD (Rank 8 and 11) were also included in the model with the highest accuracy. The inclusion of these features by the models might be because they could directly affect the margin of stability. For example, the larger trunk flexion could bring the *COM* closer to the anterior edge of the base of support, leading to an unstable or less stable *COM* state, and a higher *COM* velocity could pass the *COM* anterior to the base of support more rapidly following a trip perturbation, further worsening the *COM* state [21]. Additionally, larger trunk flexion could also increase the gravitational moments of the head–arms–trunk; therefore, a greater support impulse would be required to prevent the limb collapse towards the ground post-trip. Hence, an effective means for fall-risk reduction could be lowering the gait speed and increasing the trunk vertical orientation.

Inconsistent with previous studies [9,11,21], our results suggested that the trip-related fall risk was not simply determined by single or dual factors but jointly affected by many gait characteristics. To achieve a higher prediction accuracy (>85%), at least eight features should be involved in the fall-risk prediction models. Even though a previously developed fall-prediction model showed an accuracy of around 90% based on step time and step length, 49 trials (39 no-falls and 10 falls) were used for model training but no model validation was conducted [21]. Such a design could lead to a biased result, and the small sample size further lowered the reliability of their model. Our study used a 5-fold stratified method to split the larger sample size (*n* = 298) into a training set and test set; the test set was used to evaluate the model fit on the training set and such a design could provide an unbiased evaluation of the final model. Although step length and step time were initially included in our study for model training, they were not used to train the model as they were not selected by the feature-ranking technique. This inconsistency could possibly be derived from the inclusion of joint angles in our study, as step length could be accurately calculated based on the joint angles of the lower limbs, such as the hip and knee [33]. Rather than utilizing step length in our model, all lower-limb joint angles were incorporated (left and right hip, left and right knee). Regarding step time, which might be excluded by the feature-ranking technique due to its strong correlation with gait speed, a faster speed was found to accompany a shorter step time, and vice versa [34]. Therefore, the exclusion of these two features did not indicate that they were not related to fall risk.

In addition to the two key features (gait speed and trunk angle), toe clearance (perturbed side) and right (unperturbed side) foot angle also contributed to the trip-related fall risk prediction based on our models. Previous findings have revealed that toe clearance is an important gait parameter associated with trip-related falls [35]. Sufficient toe clearance during walking could lower the likelihood of tripping over unseen obstacles or ground irregularities, thereby decreasing the risk of trip-related falls [36]. Our previous study also found that healthy older adults increased their toe clearance in their normal walking after receiving repeated trip perturbation training [27], suggesting that increasing toe clearance is an effective strategy to avoid a trip-related fall. The foot angle of the unperturbed limb could also affect the trip-related fall risk. One of the possible reasons might be that the foot angle of the trailing (unperturbed) limb at touchdown of the leading limb might reflect the capacity of the push-off force in the ankle joint [37], and such a push-off force could directly affect the recovery response following trip perturbation, as Pijnappels et al. reported that fallers showed a slower force development in the ankle joint during push-off with the trailing limb after tripping compared to no-fallers [38].

Among all three groups, the prediction accuracy of E-fall (0–20%) was much lower than the other two groups. The imbalance in sample sizes might be one of the reasons for the low prediction accuracy of E-fall compared to other two groups, and the small sample size of the E-fall group could be due to the fact that the elevating strategy could lead to a lower fall rate than the lowering strategy [27]. Even the usage of the optimal cutoff lowered the effect of the imbalanced sample size on the prediction accuracy and improved the accuracy of E-fall prediction to around 70%, which was still much lower than the two other groups (>95% in the best model). Another possible reason for this low prediction accuracy might be that the E-fall group has a similar gait pattern to the no-fall group, and their fall risk might be due to aging-related changes in neuromuscular functions, which could not be fully reflected in gait pattern. Hence, some E-falls were falsely classified into the no-fall group and some no-falls were identified as E-falls, leading to a lower accuracy for E-fall prediction. The fall risk in the L-fall group might be related to gait patterns with a mild impairment or abnormality; hence, these fallers could be accurately predicted based on gait characteristics.

Limitations exist in our study. First, our models could be affected by the selection of feature-ranking techniques, as only the top 20 features were selected to ensure the high classification performance of the prediction model; it is possible that we may have missed certain features that could have further improved the performance of the model. Furthermore, different feature-ranking techniques (relief, mrmr, and fscnca) might result in slight differences in the selected features, while previous studies have shown that there is a limited effect of the selected features on the trained model [39,40]; hence, only the popular relief method was used in our study. Next, only kinematic factors were included in our study, as this measure could be quickly and accurately assessed without the assistance of clinicians/physicians; however, the inclusion of kinetic factors (ground reaction force) and clinical measures (timed up and go, the Berg balance scale, and the activities-specific balance confidence scale) might further improve the prediction accuracy for trip-related fall risk and this should be verified in future studies.

## 5. Conclusions

In conclusion, this study developed trip-related fall risk prediction models for healthy older adults based on their regular gait pattern, and our results indicated that machine-learning methods could accurately predict the trip outcomes with an accuracy of over 90% using an ECM algorithm. These fall-risk prediction models could easily identify individuals at a high risk of trip-related falls. Although the best model in this study could predict over 98% of fallers using a lowering strategy, only 68% of fallers using elevating strategies were accurately predicted, suggesting that the individual with a risk of L-fall could benefit more from our models compared to those with a risk of E-fall. Future work would be required to further improve the prediction accuracy for E-fall. Fall-risk detection obtained from gait pattern could be used as screening to design and implement fall-prevention training which could result in a reduction in the incidence of falls and fall-related injuries.

## Figures and Tables

**Figure 1 sensors-23-05536-f001:**
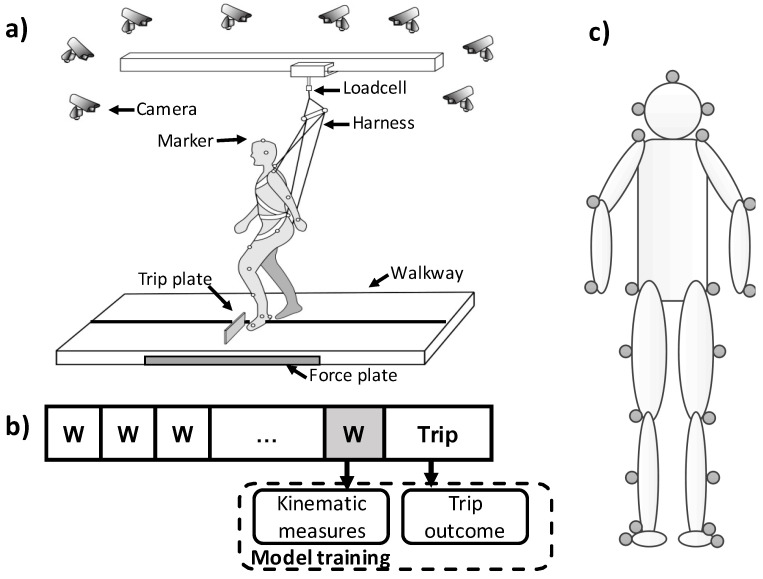
A schematic of (**a**) experimental setup, (**b**) experimental protocol, and (**c**) marker set. Subjects experienced ≥25 walking trials and 1 trip trial on the 7-m walkway. During walking trials, the trip plate was locked, and it was released during the mid-swing phase in the trip trial. In total, 26 markers were attached on the subjects; 5 of them attached on the back of the subjects are not shown in this figure, which are neck, right scapula, sacrum, right heel, and left heel markers.

**Figure 2 sensors-23-05536-f002:**
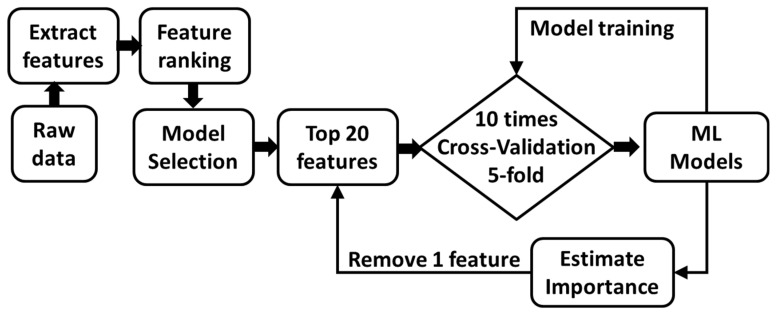
The flow chart of the model-training process.

**Figure 3 sensors-23-05536-f003:**
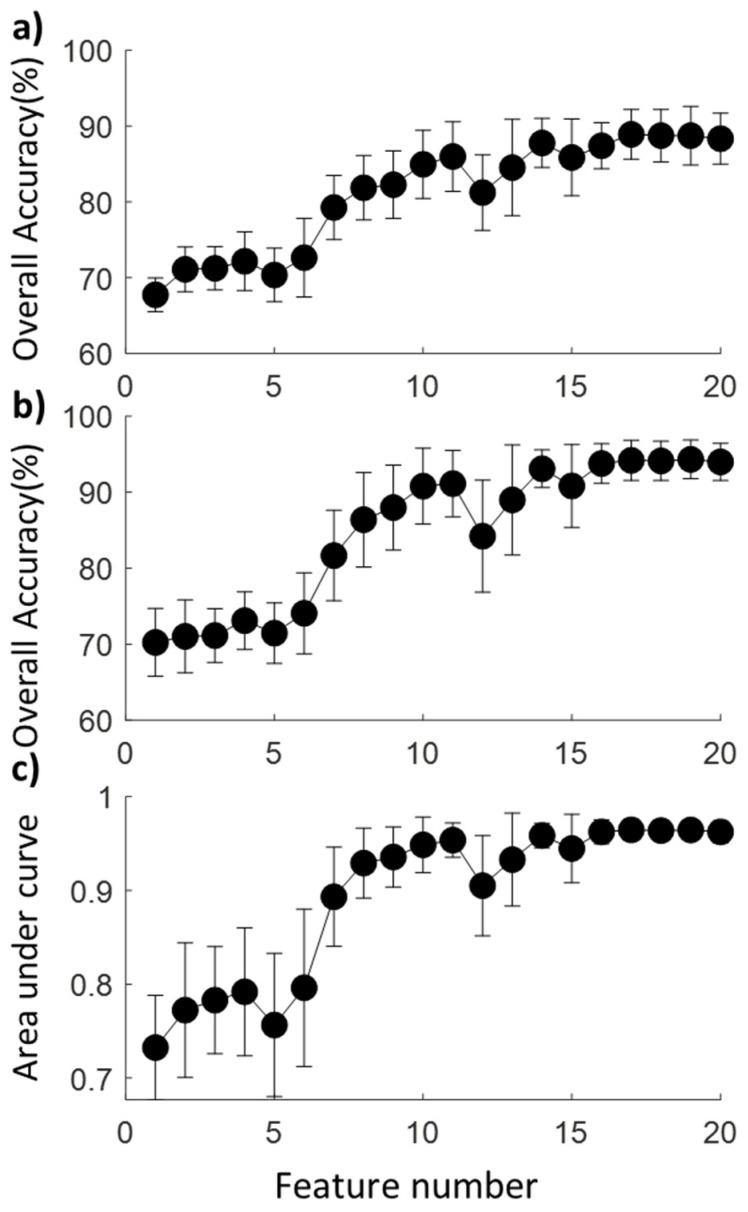
The mean and SD for (**a**) overall accuracy at the default cutoff, (**b**) overall accuracy at the optimal cutoff, and (**c**) area under the curve (AUC) for all 20 models.

**Figure 4 sensors-23-05536-f004:**
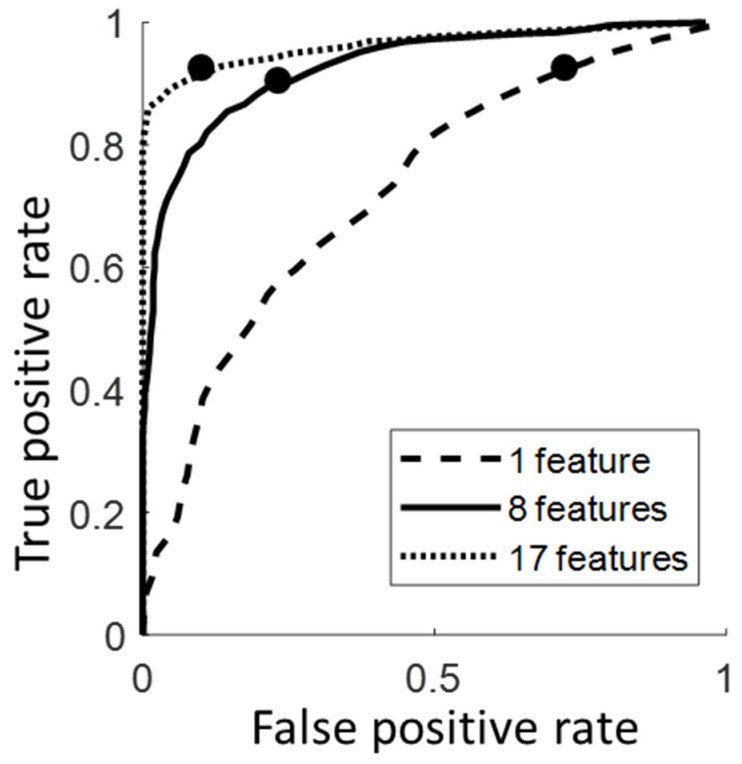
The ROC curve for the three models (a model with a single feature, the optimal model, and the best model with the highest AUC); the solid dot indicates the optimal cutoff point for each model.

**Table 1 sensors-23-05536-t001:** The mean ± SD of the top 20 features used for model training. The left limb was the perturbed limb, and the right one was the unperturbed limb.

Feature (Unit)	No-Fall	E-Fall	L-Fall
*COM* velocity at LO (m/s)	1.03 ± 0.19	1.17 ± 0.22	1.19 ± 0.54
L hip angle at Post-TD (deg)	108.08 ± 4.92	110.22 ± 3.58	109.91 ± 5.18
Gait speed in gait cycle (m/s)	1.03 ± 0.19	1.17 ± 0.21	1.12 ± 0.19
Max knee flexion in swing phase (deg)	119.18 ± 9.15	116.76 ± 6.62	116.50 ± 10.21
R foot angle at post-TD (deg)	−9.09 ± 7.11	−12.23 ± 8.55	−12.32 ± 8.33
L knee angle at post-TD (deg)	178.15 ± 8.80	176.36 ± 8.23	177.07 ± 8.97
Toe clearance (m)	0.16 ± 0.03	0.16 ± 0.02	0.16 ± 0.02
Max trunk flexion in gait cycle (deg)	86.08 ± 5.34	84.88 ± 6.48	86.67 ± 5.70
*COM* velocity at pre-TD (m/s)	1.06 ± 0.21	1.20 ± 0.24	1.16 ± 0.20
R hip angle at post-TD (deg)	74.05 ± 6.46	71.71 ± 6.32	72.54 ± 4.46
Trunk angle at post-TD (deg)	89.23 ± 4.73	88.11 ± 5.91	89.87 ± 5.89
Max hip flexion in swing phase (deg)	112.44 ± 5.11	112.86 ± 3.98	114.19 ± 5.65
R knee angle at post-TD (deg)	171.79 ± 8.47	173.67 ± 9.50	172.08 ± 7.45
L hip angle at LO (deg)	87.71 ± 6.05	87.15 ± 7.00	86.05 ± 6.17
Max gait speed in gait cycle (m/s)	1.11 ± 0.20	1.25 ± 0.23	1.28 ± 0.63
R knee angle at LO (deg)	164.41 ± 9.24	162.50 ± 9.37	164.37 ± 9.36
L foot angle at post-TD (deg)	15.68 ± 7.12	17.54 ± 5.84	16.81 ± 6.27
R hip angle at LO (deg)	102.18 ± 5.75	104.43 ± 6.01	103.60 ± 5.68
Trunk angle at LO (deg)	88.57 ± 4.80	87.15 ± 6.25	89.12 ± 5.62
L knee angle at LO (deg)	134.20 ± 10.42	131.71 ± 10.11	133.72 ± 10.32

**Table 2 sensors-23-05536-t002:** Prediction accuracy at the default cutoff for all 20 models containing different numbers of features (feature number: 1–20); each model contains all the features listed in the corresponding row and above. The prediction accuracy (true positive rate) for each trip outcome and the overall prediction accuracy were calculated. The left limb was the perturbed limb, and the right one is the unperturbed limb.

Feature Number	New Feature	Overall Accuracy			
		Overall	No-Fall	E-Fall	L-Fall
1	*COM* velocity at LO	67.7	97.1	0.0	18.5
2	L hip angle at post-TD	71.1	94.1	0.0	37.3
3	Gait speed in gait cycle	71.2	94.9	0.0	35.9
4	Max knee flexion in swing phase	72.2	96.5	0.0	35.6
5	R foot angle at post-TD	70.4	96.5	0.0	29.2
6	L knee angle at post-TD	72.6	96.6	0.5	36.8
7	Toe clearance	79.3	98.6	2.5	55.4
8	Max trunk flexion in gait cycle	81.9	99.3	6.9	62.0
9	*COM* velocity at pre-TD	82.3	99.2	7.6	63.3
10	R hip angle at post-TD	85.0	99.5	11.5	71.1
11	Trunk angle at post-TD	86.0	99.6	15.6	73.6
12	Max hip flexion in swing phase	81.2	99.3	3.3	60.5
13	R knee angle at post-TD	84.5	99.4	8.3	70.8
14	L hip angle at LO	87.8	99.6	14.7	80.1
15	Max gait speed in gait cycle	85.9	99.5	13.1	73.9
16	R knee angle at LO	87.4	99.5	11.4	80.0
17	L foot angle at post-TD	88.9	99.8	17.8	83.0
18	R hip angle at LO	88.7	99.9	20.0	81.5
19	Trunk angle at LO	88.7	99.9	17.2	82.3
20	L knee angle at LO	88.3	99.5	17.1	81.9

**Table 3 sensors-23-05536-t003:** Prediction accuracy at the optimal cutoff for all the 20 models containing different number of features (feature number: 1–20); each model contains all the features listed in the corresponding row and above. The prediction accuracy (true positive rate) for each trip outcome and the overall prediction accuracy were calculated. The left limb was the perturbed limb, and the right one is the unperturbed limb.

Feature Number	New Feature	Overall Accuracy			
		Overall	No-Fall	E-Fall	L-Fall
1	*COM* velocity at LO	70.2	87.5	0.0	49.3
2	L hip angle at post-TD	71.0	82.1	0.0	64.6
3	Gait speed in gait cycle	71.2	82.1	0.0	64.9
4	Max knee flexion in swing phase	73.1	87.0	0.0	60.5
5	R foot angle at post-TD	71.5	84.4	0.0	60.7
6	L knee angle at post-TD	74.1	86.1	3.2	65.2
7	Toe clearance	81.7	90.5	16.3	78.6
8	Max trunk flexion in gait cycle	86.4	90.9	38.9	88.4
9	*COM* velocity at pre-TD	88.0	92.5	40.4	90.2
10	R hip angle at post-TD	90.8	94.1	50.5	93.9
11	Trunk angle at post-TD	91.1	93.3	56.6	95.3
12	Max hip flexion in swing phase	84.2	91.8	23.9	82.5
13	R knee angle at post-TD	89.0	93.7	42.9	90.2
14	L hip angle at LO	93.1	94.6	60.1	98.2
15	Max gait speed in gait cycle	90.8	93.9	52.2	94.1
16	R knee angle at LO	93.8	95.5	62.7	97.9
17	L foot angle at post-TD	94.2	95.3	68.2	98.7
18	R hip angle at LO	94.1	95.0	68.2	99.0
19	Trunk angle at LO	94.3	95.3	69.7	98.6
20	L knee angle at LO	94.0	94.9	68.4	98.7

## Data Availability

The data presented in this study are available on request from the corresponding author. The data are not publicly available due to privacy issue.

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
