# Peer review of "Trip-Related Fall Risk Prediction Based on Gait Pattern in Healthy Older Adults: A Machine-Learning Approach"

_sensors, 2023, doi:10.3390/s23125536_

Round 1

Reviewer 1 Report

1. Experimental measurements should be added in the revised paper.

2. Original signals of three classes should be provided in the revised paper for clear understanding of gait patterns

3. There is no any description of calculating equations of different features.  

Reviewer 2 Report

A novel and meaningful study was conducted. There are 2 suggestions to improve the paper.

1.    The paper concerned the trip-related fall risk of elderly people, that should be clearly expressed in the titled.

2.    In the second part of paper, the experimental setup was not clearly illustrated, a sketch gram of the experimental setup is recommended.

Reviewer 3 Report

The word "while" was used unnecessarily a couple of times and it made sentences confusing. Intro paragraph 1 and next to last paragraph of discussion. 

I am a basic scientist and could not really envision your design. A methods figure would be a significant addition to the paper. A useful figure might show the arrangement of sensors on a stick figure or drawing. This could also be referenced when describing calculations made later on this page. 

Figure 1 text: delete first "to" in "...were also considered to related to fall..."

In your results I had to make some assumptions. Table 1 shows increasing overall accuracy with increased feature number. Does Feature 17 data contain 17 features, or a single feature? This is not clear.

You use "feature" and "model" interchangeably, and unless I am missing something, it might be beneficial to readers if you stick to "feature" unless you are referring to a combination of features that produce a predictive model. This was very hard to follow for me and could probably benefit from some rewording and proofreading.

Please spell out TUG, BBS, ABC clinical measures for clarity.

I appreciate the discussion about your E-fall group, but still feel like this group is being ignored in your conclusions that this is a good model for predicting falls in general. This model shows a great deal of promise for a specific type of faller, but falls short for fallers that make it past the obstacle. In a nutshell, I think that your conclusions can be more specific. You have room to describe the type of faller that will benefit from your model.

N/A

Round 2

Reviewer 1 Report

The revised paper can be  accepted for publication

Author Response

Thank you for your positive feedback!